# Maternal Plasma Glycerophospholipids LC-PUFA Levels Have a Sex-Specific Association with the Offspring’s Cord Plasma Glycerophospholipids-Fatty Acid Desaturation Indices at Birth

**DOI:** 10.3390/ijerph192214850

**Published:** 2022-11-11

**Authors:** Sowmya Giriyapura Vamadeva, Nagalakshmi Bhattacharyya, Kunal Sharan

**Affiliations:** 1Department of Molecular Nutrition, CSIR-Central Food Technological Research Institute, Mysuru 570020, India; 2Academy of Scientific and Innovative Research (AcSIR), Ghaziabad 201002, India; 3Kamakshi Hospital, A Unit of B.S.M.S Trust (Regd), Mysuru 570009, India

**Keywords:** pregnancy, LC-PUFA, essential fatty acids, fatty acid desaturases, sexual dimorphism, fetal programming

## Abstract

Fatty acid desaturases, the enzymes responsible for the production of unsaturated fatty acids (FA) in fetal tissues, are known to be influenced by maternal-placental supply of nutrients and hormones for their function. We hypothesize that there could be a gender-specific regulation of unsaturated FA metabolism at birth, dependent on the maternal fatty acid levels. In this study, 153 mother-newborn pairs of uncomplicated and ‘full-term’ pregnancies were selected and the FA composition of plasma glycerophospholipids (GP) was quantified by gas chromatography. The FA composition of mother blood plasma (MB) was compared with the respective cord blood plasma (CB) of male newborns or female newborns. Product to substrate ratios were estimated to calculate delta 5 desaturase (D5D), delta 6 desaturase (D6D) and delta 9 stearoyl-CoA-desaturase (D9D/SCD) indices. Pearson correlations and linear regression analyses were employed to determine the associations between MB and CB pairs. In the results, the male infant’s MB-CB association was positively correlated with the SCD index of carbon-16 FA, while no correlation was seen for the SCD index of carbon-18 FA. Unlike for males, the CB-D5D index of female neonates presented a strong positive association with the maternal n-6 long chain-polyunsaturated FA (LC-PUFA), arachidonic acid. In addition, the lipogenic desaturation index of SCD18 in the CB of female new-borns was negatively correlated with their MB n-3 DHA. In conclusion, sex-related differences in new-borns’ CB desaturation indices are associated with maternal LC-PUFA status at the time of the birth. This examined relationship appears to predict the origin of sex-specific unsaturated FA metabolism seen in later life.

## 1. Introduction

Pregnancy is an adaptive physiology, which supports growth of the fetus in a nutrient-rich environment [1]. Nutrients like essential fatty acids (EFA) are acquired through diet, as they are not synthesized by humans. Dietary availability of EFA-alpha-linolenic acid (ALA), an omega-3 (n-3) polyunsaturated fatty acid (PUFA) precursor, and linoleic acid (LA), an omega-6 (n-6) precursor, is essential. These n-6 and n-3 fatty acids (FA) are substrates for the same fatty acid desaturases and elongases, in order to convert them to the respective long-chain PUFAs (LC-PUFA) [2]. Arachidonic acid (n-6, ARA), Eicosapentaenoic acid (n-3, EPA) and Docosahexaenoic acid (n-3, DHA) are the long-chain derivatives of n-6 and n-3 precursors and are essential components of cell membranes [3]. The functional roles of n-6 and n-3 LC-PUFA derivatives are opposite in nature, as they possess pro- and anti-inflammatory properties, respectively. Thus, a balanced n-6/n-3 ratio in the maternal diet plays an indispensable role during pregnancy and lactation [4,5,6,7]. The daily recommendation for n-3 LC-PUFA intake is 500 mg as n-3 DHA and n-3 EPA, or 1–2 servings of fish per week for an adult. During pregnancy, LC-PUFAs potentiate the constitution of vital organs in the fetus [8]. Hence, an early n-3 DHA enrichment happens in the maternal plasma [9]. Moreover, high requirements of LC-PUFA in fetal organ development drive an enhanced surge of LC-PUFAs in the cord blood, in divergence from maternal blood, by a phenomenon called ‘biomagnification’ [10,11,12].

A global increase in maternal n-6/n-3 PUFA ratios has caused altered metabolic outcomes at the birth of offspring [13,14] and in their later adult life [6,7,15,16]. Additionally, the role of early EFAs/LC-PUFAs in the programming of fatty acid desaturases, the enzymes responsible for the desaturation of FA, is a matter of debate. Delta-5-desaturase (D5D) and delta-6-desaturase (D6D) are a set of critical enzymes genetically coded by *FADS1* and *FADS2* genes which is responsible for endogenous synthesis of LC-PUFAs [17] and is affected by the type of PUFA diet. Another enzyme, delta-9 stearoyl-CoA-desaturase (SCD1) targets the desaturation of saturated fatty acids (SFA) to monounsaturated fatty acids (MUFA) and is regulated by PUFA synthesis and availability of PUFA precursor substrates [18].

Human and animal fatty acid desaturases are expressed in a variety of tissues [19] and their regulation exhibits a prominent sexual dimorphism. In adults, a gender-dependent endogenous synthesis of LC-PUFAs from parent precursors is favored by sex hormones, and is more significant in women of reproductive age than men [20,21,22]. In addition, numerous animal studies have demonstrated promotion by estrogen of LC-PUFA biosynthesis in adult females with parent precursor supplementation, which is less efficient in adult males [23,24]. Interestingly, estrogen downregulates SCD1 mRNA expression in adipose tissue [25,26]. Moreover, a human study observed a sex-related high total MUFA content in healthy men and a high n-6 LA with consequent higher total n-6 LC-PUFA in healthy women subjects [27]. 

The above-mentioned studies provide important information regarding the gender differences in the metabolism of unsaturated FA at adult age. However, there is no data about the sex-specific in utero LC-PUFA status in terms of estimated/surrogate desaturation indices in healthy full-term newborns, particularly in a population dominated by vegetarians. Hence, we hypothesized that there could be causal relationships between maternal LC-PUFA levels and offspring’s CB-FA desaturation indices at birth in a sex-dependent manner. We examined these relationships for a gender-specific grouping of healthy new-borns and their mothers, who consumed a diet containing high n-6 LA from vegetable oils and inadequate n-3 LC-PUFA during pregnancy.

## 2. Methods and Materials

### 2.1. Human Observation Study Design

All subjects gave their informed consent for inclusion before they participated in the study. The study was conducted in accordance with the Declaration of Helsinki, and the protocol was approved by the Apollo BGS Hospital Ethics Review Board (ECAN-16/11 April 2018), Mysuru, India. Blood was sampled from healthy pregnant women admitted to the maternity unit of Kamakshi hospital, Mysuru, at the time of delivery. The study samples were collected from August 2018 to November 2018. A total of 232 mother and cord blood pairs were collected. After reviewing medical records, 153 pairs were found eligible in terms of the inclusion and exclusion criteria set for the study. 

An oral survey using a 24-h recall method was done to assess the dietary pattern of maternal subjects before delivery (last month of the pregnancy). The oral survey included a general nutritional questionnaire and qualitative (relative) information about the diet to estimate frequency, amount and type of food consumed for the previous 24 h. We also collected information about the frequency of fish or any other health supplement intake during pregnancy. Based on the questionnaire, the nutritional contents and dietary intake values (%) of the respondent’s diet was calculated using the Indian Food Composition Database (IFCT 2017). The levels of essential fatty acids (PUFA) and their metabolites in the diet were also estimated. Nutritional data obtained was recorded in a spreadsheet. 

Figure 1 outlines the design of a human cross-sectional study. The following criteria were considered for the random selection of samples for the study. 

Inclusion criteria: Healthy pregnant women with ‘full-term’ deliveries, and no previous medical illness history/complications. Only singleton deliveries were included.

Exclusion criteria: preterm and twin deliveries, gestational diabetes, pregnancy-induced hypertension (PIH), thyroid problems, cases of maternal anemia, physiological jaundice in newborn, low birth weight (<2500 g). The samples were first collected then excluded according to the exclusion criteria. The subjects were made aware of this as a possibility.

### 2.2. Quantification of Fatty Acids in Plasma Glycerophospholipids of the Mother and Cord Blood Pairs by Gas Chromatography

Venous blood from the mother and double-clamped maternal-fetal interface of the umbilical cord was collected in EDTA tubes for plasma separation at the time of delivery using a syringe needle. Samples were centrifuged at 3600 rpm to collect the upper phase plasma in another tube and stored at −80 °C until analyzed. The plasma GP fatty acid composition was measured using a sensitive base catalyzed transesterification method, equivalent to the phospholipid fraction for precise high-throughput analysis as described by Glaser et al. [28]. Briefly, sodium methoxide was used for the derivatization of fatty acids to fatty acid methyl esters (FAME). FAME were separated by employing the previously described protocol using a restek Rtx-2330 (fused silica) column and quantified by Shimadzu gas chromatography (GC-2010 plus model) with flame ionization detection. Individual fatty acids were identified by comparison of retention time with an authentic FAME standard mixture. Integral GC solution software was used for peak integration. The results are expressed as percentage (%) of each FAME by weight of all GP-FAME detected. FAs were considered undetected when a peak was found too low for integration. The status in percentage terms of the following fatty acids was derived from each sample: total SFA (sum of C14, C16 and C18), total MUFA (C16:1 and C18:1), and total n-3 PUFA (C20:5, C22:5 and C22:6), total n-6 PUFA (C18:2, C20:3 and C20:4) and total n-6/n-3 ratio. 

### 2.3. Quality Controls of Gas Chromatography Analysis

An external standard was analyzed after every batch of ten samples on the GC/FID to verify the precision of the FA calibration curves. If the measured percentages of these standards differed by more than 5 percent, the samples were reanalyzed for operational specifications. Repeatability was evaluated in an inter-days and same-day basis by validating the co-efficients of variation (CV), <2%, of individual samples. Solvent blanks (to assess the clean status of the equipment) and reagent blanks with water as sample (to assess background-noise in the chromatogram) were prepared for each batch of samples injected in a day. Samples were trans-esterified on the day of GC analysis to maintain sample stability and avoid errors during storage. The transesterification efficiency of this method was validated and established by Glaser et al. [29,30] and others [31].

### 2.4. Desaturation Indices Calculation

The desaturation indices were estimated as reported earlier [13,16] using the FA product/substrate (P/S) ratio of the enzymes in the following way: D9D or SCD1 index for C16 and C18 FA precursors- SCD16 (16:1n7/16) [13], SCD18 (18:1n9/18) [13], D5D (20:4n-6/20:3n-6) [16] and D6D (20:3n-6/18:2n-6) [16].

### 2.5. Statistical Analysis

Means and standard deviations were used to describe data variables. Data variables followed a normal distribution as tested by the Shapiro–Wilk test. Pearson’s correlation (r) was determined by linear regression analysis using GraphPad Prism, v8.0. Adjusted *t*-tests were performed for pairwise comparison between groups. A *p*-value < 0.05 was considered statistically significant.

## 3. Results

### 3.1. Study Subjects

The clinical characteristics of the study participants are given in Table 1. No significant contribution of maternal gestational age and gender-specific newborn body weights were found towards the study outcome. According to nutritional recall oral survey, some maternal subjects were non-habitual consumers of n-3 DHA in their diet (through fish or supplements). All the subjects consumed vegetable oil as a source of high n-6 LA with no smoking or alcoholic history during gestation or before, as recorded during the 24 h diet recall oral survey. There was no significant difference in the dietary fat intake between maternal groups bearing male or female offspring (Appendix A). 

### 3.2. Comparisons of Maternal and Sex-Specific Newborn’s Cord Blood Fatty Acid Composition at Birth

Table 2 shows the levels of FA measured in the MB and CB plasma GP in a heatmap for the studied subjects. The most abundant FAs in maternal plasma GP were C18:1 and C18:2 n-6 FA. ARA was the other significant n-6 LC-PUFA with a concentration of 7% of the total FA. There was a decrease in the n-6 PUFA precursor LA of the male and female CB when compared to the MB. However, female CB ARA levels were significantly higher than levels in the male CB. A significant decrease in the n-6/n-3 ratio in male and female CB was found, as compared to MB levels. The total MUFA levels were significantly decreased in the CB of both genders when compared with the respective MB. Figure 2 shows that higher average index values SCD16, D5D and D6D were found in the CB of both newborn genders compared to MB. The values of the CB-D5D index for female neonates were significantly higher than for male infants. There was also a decrease in the SCD18 index of CB in both genders when compared to MB.

### 3.3. Relationships between Fatty Acids of Maternal and Cord Blood Plasma Glycerophospholipids in Male and Female New-Borns

Figure 3 shows the heatmap summary of correlation coefficients between MB-CB FA in male and female new-borns at the time of birth. Levels of maternal SFA-C14, C16, and MUFA-C18:1 were significantly positively correlated with those of the corresponding fatty acids in CB of both males and females. However, unlike for male CB, female CB MUFA-C16:1 and SFA-C18 presented a significant positive correlation with the maternal levels. The levels of n-6 PUFA, LA, and DHGLA showed a significant positive correlation between MB and CB of new-borns of both sexes. In addition, LC-PUFA n-6 ARA and n-3 DHA showed a positive correlation between MB and CB of male and female neonates. Total SFA, total MUFA, and total PUFA showed strongly significant correlations between MB and CB of female infants collected at birth. Interestingly, in contrast to males’ CB, female CB showed a significant correlation with maternal n-3 DPA and total SFA. Strong positive correlations were observed between MB and newborn’s CB of both sexes for total MUFA, total PUFA, total n-3 total n-6, and total n-6/n-3 ratio (Figure 3). Correlation coefficients between each corresponding FA of MB with the respective FA of CB plasma GP in male and female infants is detailed in Appendix A. 

### 3.4. Associations of Maternal and New-Born’s Sex-Specific Cord Blood Desaturation Indices at Birth

To examine the sex-specific desaturase activities at birth and their association with corresponding maternal desaturation index, we evaluated MB-CB desaturase activities (P/S ratio) by Pearson correlation and regression analyses for both newborn gender groups. Male offspring’s CB SCD16 desaturation index showed a significant association with their respective maternal SCD16 index (Figure 4A). However, the male CB SCD18 was weakly related to maternal SCD18 (Figure 4B). In contrast, CB SCD16 and CB SCD18 of females presented a strong positive association with maternal SCD16 and SCD18 index (Figure 4E,F). Further, the association of both genders’ CB D5D and D6D with maternal D5D and D6D was statistically significant and showed a positive correlation (Figure 4C,D,G,H).

### 3.5. Associations of Maternal LC-PUFA with the Newborn Gender-Related Cord Blood Desaturation Indices

To ascertain whether maternal LC-PUFA status is associated with sex-specific differences in desaturation index, maternal PUFA status was assessed with a total n-6/total n-3 ratio as the starting point. There was a significant positive correlation between maternal PUFA status and CB SCD16 index of male and female infants (Figure 5A,D). The relationship of maternal PUFA status was negative with the CB D6D of infants of both genders (Figure 5C,F). However, CB SCD18 was positively associated with the maternal PUFA status in female newborns only, with no significant association with the males’ CB (Figure 5B,E). To test whether maternal n-3 or n-6 LC-PUFA was related to the cord blood desaturation index in newborn gender groups, we performed a correlation analysis between the sex-specific CB desaturation index and maternal DHA or ARA status. Maternal n-6 ARA and n-3 DHA showed no significant correlations with CB SCD16 and CB SCD18 in males (Figure 6A,B). However, in female infants, CB SCD16 presented a significant strong negative association with maternal ARA, and CB SCD18 had a negative correlation with maternal DHA levels (Figure 6E,F). Maternal DHA showed a strong positive association with CB D6D of both gender groups (Figure 6D,H). In contrast to a weak correlation of male CB-D5D with MB ARA, female infants indicated a significant positive association (Figure 6C,G). 

## 4. Discussion

In the present study, associations between MB LC-PUFA and venous CB desaturation indices at the time of birth confirm our hypothesis, showing significantly different relationships for male and female offspring. The study was conducted in a population dominated by vegetarians with low access to dietary n-3 LC-PUFA. Numerous reports suggest that the in utero environment and nutrition are linked to several diseases at adulthood of the offspring [32,33,34]. Interestingly, there is a gender difference in the way a fetus adapts to the in utero nutritional changes [35,36]. Sex-specific programming during gestation holds metabolic cues for the offspring’s later life [37]. Sexual dimorphism in the levels and metabolism of LC-PUFA in adults have been reported earlier [22]. However, the preference for distinct levels of LC-PUFA might be programmed early during fetal sex development and later influenced by the sex hormones [38]. Even during fetal development, a sexually dimorphic placental transcription occurs in response to n-3 LC-PUFA supplementation [39]. The current study for the first time reports the role of the fetal gender in utero in differential metabolism of LC-PUFA, measured at birth. It also produces cues for design of future studies in the area with emphasis on sex-differences in early life.

The placenta and its extended cord serve as a dominant interface that controls a multifaceted interplay between maternal-fetal lipid metabolic interactions [40], which are regulated by hormones of maternal and sex-specific fetal origin [36,41]. Previous evidence suggests the importance of the maternal supply of EFAs, especially LC-PUFAs, in fetal growth and development in utero [42]. However, studies focusing on the effect of maternal LC-PUFAs on offspring’s PUFA metabolism at birth are uncommon. We observed a ‘biomagnification’ of LC-PUFAs in the offspring, which is in line with previous studies reporting increased levels of LC-PUFAs in the venous CB, which occurs to meet the high demands of the growing fetus [10]. However, data from studies of complicated gestation suggest a link between disturbed maternal hormonal milieus and reduced placental transfer of LC-PUFAs, impacting fetal growth and birth outcomes adversely [36,43]. 

FA desaturases carry out desaturation, a crucial process for the generation of LC-PUFAs, which are precursors of pro-inflammatory and anti-inflammatory eicosanoids [44]. The functions of the FA desaturases are triggered by the growing needs of fetal development across pregnancy, the maternal environment of FA precursor levels and placental transfer of LC-PUFAs to the fetus [45]. Despite reports confirming the programming effects of in utero nutritional modulations in the fetus, understanding of how maternal LC-PUFAs relate with gender-dependent desaturases in the intrauterine environment remains elusive. In this study, we observed a sexual dimorphic trend in the FA profiles of MB-CB pairs from male and female full-term newborns. The LC-PUFA relationships in the current study point towards an unmapped differential PUFA metabolism in human fetuses for the first time. This phenomenon may be affected by the complexity of the maternal hormonal milieu system and its response to placental dimorphism in relation to fetal sex.

Human adults are capable of LC-PUFA biosynthesis, affected by a sexually dimorphic influence of steroid sex hormones exercising regulatory functions on FA desaturases. However, there are no reports on the associations of maternal LC-PUFA levels with the sex-specific CB fatty acid desaturation indices involved in the human metabolism of the unsaturated fatty acids at birth. The activity of SCD (a MUFA desaturase) is known to have an inverse relationship with PUFA status [46]. Thus, we aimed to assess SCD desaturation, often viewed as a lipogenic index and a valid metabolic marker of adiposity. Our study showed a distinctive suppression of the SCD18 (oleic acid/stearic acid) index in the CB of both male and female newborn groups, as a result of high PUFA-D5D and D6D desaturation indices of the CB at birth (Figure 3). Interestingly, the correlations of desaturation indices were sexually dimorphic for MB-CB SCD16 and MB-CB SCD18 relationships (Figure 5). Unlike the MB-CB SCD indices of females, only MB-CB SCD16 was positively correlated in male newborns. This may relate to gender-related early accumulation of adipose tissue in females, as a long term adaptation, whereby the physiology of adult females accounts for greater fat composition than males [47,48]. Here, the SCD index may be an early marker of sex difference in whole body composition being programmed in utero. Moreover, MB DHA and MB ARA showed no significant correlation with CB SCD16 and CB SCD18 in males and a significant negative association in females. This could explain sex-specific consequences for differential regulation of 16-carbon and 18-carbon FA desaturation pathways [6,46,49]. In addition, recent identification of palmitoleate as a lipokine [50] and main regulator of SCD1 expression may substantiate the function of SCD1 in fetal sex development; however, a study has demonstrated the contribution of fetal sex in high placental uptake of oleic acid and arachidonic acid in obese pregnant women [14]. To the best of our knowledge, there are no related studies reporting relationships between maternal-fetal genders in normal-healthy pregnancies. We and others [39] believe ‘sex/gender’ of the newborn is one of the possible biological explanations of the MB-CB analyses. This programming effect of metabolic pathways in the sexually dimorphic CB evaluated in terms of desaturation indices could reflect an influence of fetal sex on the placenta.

An imbalanced or a higher ratio of maternal n-6/n-3 PUFA is often correlated with increased measures of adiposity in growing animals (6) and human offspring [51]. In this study, we found a positive correlation between maternal n-6/n-3 PUFA ratios and CB SCD16, which was unaffected by the newborn’s sex (Figure 6). This intertwined yet opposite relationship of MUFA-lipogenic SCD1 and PUFA-D5D and D6D desaturases with the maternal levels of n-3 and n-6 LC-PUFA may perhaps program sex-specific differences in FA metabolism of the offspring in later life. These relationships indicate the complex sex-specific metabolic programming of the fetus in the maternal environment. Some earlier reports have demonstrated that fetal sex is a strong determinant of the levels of maternal hormones during pregnancy [52,53,54]. Differences in sex-hormone levels of maternal estrogen and testosterone were seen in pregnancies bearing term male and female fetuses [41]. A study by Sedlmeier et al. reported a reduced ratio of Estradiol-17β to testosterone in the placentas bearing females not males, upon maternal n-3 LC-PUFA intervention [39]. Of note, estrogen promotes D5D and D6D desaturases and testosterone inhibits the enzyme activities in animals and human adults [25]. The mechanism(s) underlying this association is still unclear and future studies are required to shed more light on it. Here, we report that sex-related differences in desaturases of unsaturated FA metabolism may have their origins and function influenced by maternal LC-PUFA status at birth. 

The strength of this study lies in the use of newborn’s sex as an important factor of study and measuring CB product-to-precursor FA as estimated desaturation indices in healthy term pregnancies. This method provides both endogenous and exogenous FA status in the healthy newborns and the relationship with maternal dietary intake status. This population study mainly included vegetarian mothers consuming n-6 LA rich vegetable oils and the diet of the population in this particular region is deficient in n-3 PUFA ALA and DHA [55,56]. Our oral survey with the maternal subjects also confirmed the lack of awareness about healthy fats in the diet.

However, the study was limited by a small-sized population with samples from one hospital. Another limitation of the study was that the desaturation indices were not measured directly but were based on estimated product-to-precursor ratios. The levels of important FAs such as n-3 ALA and n-6 GLA were too low for detection and integration in this sample study. In this study group, we also observed a high n-6 LA and very low n-3 ALA (parent precursors of LC-PUFA) diet (Appendix A). Coherent with our observation, a recent study in the south Indian population also noticed a deficiency in ALA in maternal dietary intake [55]. Further, the mothers were predominantly vegetarian and were not able to consume n-3 PUFAs up to the levels recommended by experts. Therefore, the results from a population reaching the dietary recommendation might differ from our study. 

## 5. Conclusions

In conclusion, the results of our study present a complex sex-specific association of newborns’ unsaturated FA desaturation indices with maternal LC-PUFA status in normal full-term pregnancies. These gender-related relationships between MB vs CB desaturases at birth may tend to drive the regulation of fatty acid metabolism seen in male and female adults under the influence of sex hormones. It also points towards the influence of in utero maternal LC-PUFA in alteration of sexually dimorphic FA metabolism at birth. Although our data showed that newborn groups registered a higher percentage of n-3 DHA and n-6 ARA (LC-PUFA) in the CB, only the CB-D5D index of female newborns showed an association with maternal LC-PUFA status; this was not seen for the males. This observation reflects a probable role of fetal sex and its distinct metabolic responses in the maternal environment. This interaction between maternal LC-PUFA diet and fetal sex could have an important physiological function in the growth and development of the offspring in utero. However, future investigations are required to explore the mechanisms of MB and sex-specific CB associations with maternal LC-PUFA levels and interventions.

## Figures and Tables

**Figure 1 ijerph-19-14850-f001:**
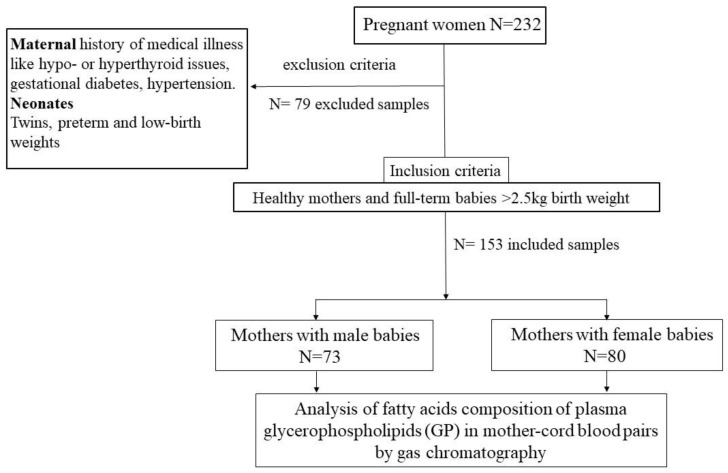
Flowchart depicting the design of the human observational study.

**Figure 2 ijerph-19-14850-f002:**
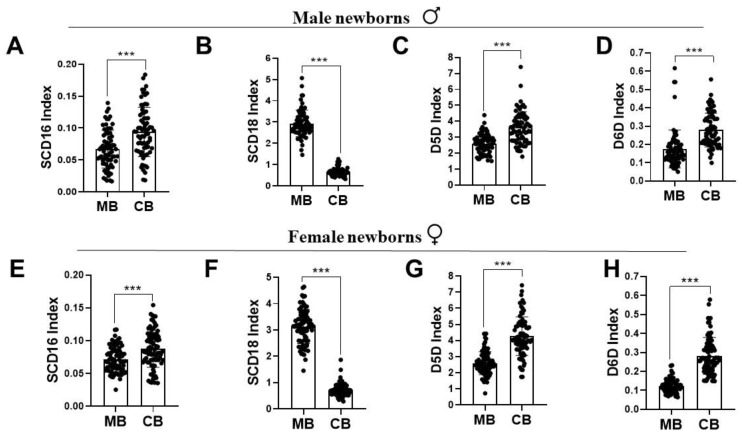
Bar graphs representing average estimated fatty acid desaturation indices (ratio of product by precursor fatty acid) from plasma fatty acid composition quantified by gas chromatography of MB and CB in the human cross-sectional study. (**A**–**D**) SCD desaturation indices of carbon 16 and 18 as SCD16 (C16:1/C16), SCD18 (C18:1/C18), delta-5-desaturases (C20:4/C20:3, D5D) and delta-6-desaturases (C20:3/C18:2), D6D) indices in the MB and CB of male newborns. (**E**–**H**) SCD16, SCD18, D5D and D6D indices in MB-CB of female newborns. MB < CB for SCD16, D5D and D6D indices and MB > CB for SCD18 index in the male and female newborns. *** *p* < 0.001 indicates significance between the compared groups. MB, plasma of mother blood; CB, plasma of cord blood; SCD, stearoyl CoA desaturase; D5D, delta-5-desaturase; D6D, delta-6-desaturase.

**Figure 3 ijerph-19-14850-f003:**
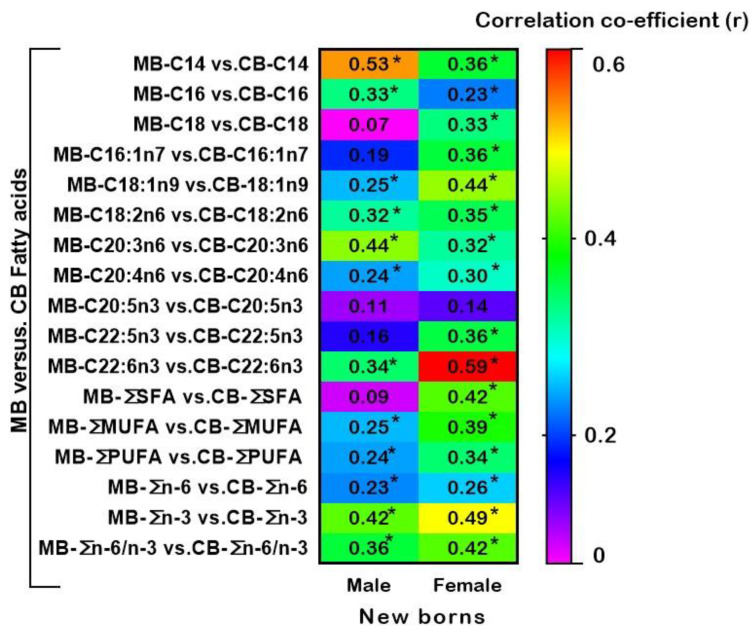
Heatmap of correlation co-efficients between individual fatty acids (%) of MB and CB at birth. A strong relationship exists between MB versus CB C22:6n3 (DHA) of female newborns. * *p* < 0.05 between compared MB and CB fatty acids. See also Appendix A for details. Abbreviations: C14:0(MA), myristic acid; C16:0(PA), palmitic acid; C18:0(SA), stearic acid; C16:1n7(POA), palmitoleic acid; C18:1n9(OA), oleic acid; C18:2n6(LA), linoleic acid; C20:3n6(DHGLA), di-homo gamma linolenic acid; C20:4n6(ARA), arachidonic acid; C20:5n3(EPA), eicosapentaenoic acid; C22:5n3, docosapentaenoic acid; C22:6n3(DHA), docosahexaenoic acid; MB, plasma of mother blood; CB, plasma of cord blood.

**Figure 4 ijerph-19-14850-f004:**
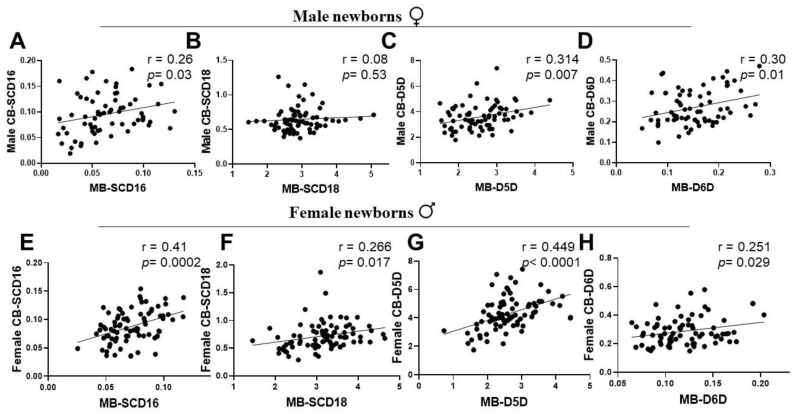
Cord blood fatty acid desaturation indices are positively associated with maternal fatty acid desaturation indices. (**A**–**D**) SCD16, SCD18, D5D and D6D indices of MB versus CB desaturation indices of male newborn group. (**E**–**H**) SCD16, SCD18, D5D and D6D indices of MB versus CB desaturation indices of female newborn group. MB versus CB correlations for SCD16, D5D and D6D indices are significantly positive. For SCD18 index, only MB and CB of female newborns are positive and remarkably significant (however, there was no relationship between MB versus CB SCD18 of male newborns). SCD, stearoyl CoA desaturase; D5D, delta-5-desaturase; D6D, delta-6-desaturase. MB, plasma of mother blood; CB, plasma of cord blood.

**Figure 5 ijerph-19-14850-f005:**
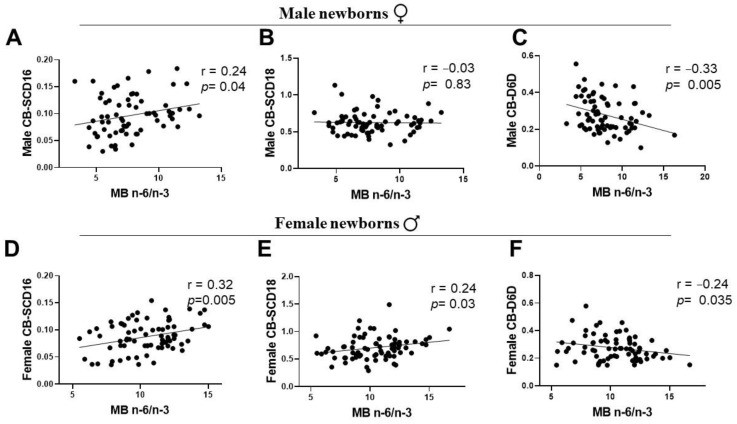
Cord blood desaturation indices are differently related to maternal total n-6/n-3 ratio PUFA status in the newborn gender groups. (**A**–**C**) Male newborns. (**D**–**F**) Female newborns. Pearson *r* and *p* values are shown. PUFA, polyunsaturated fatty acid.

**Figure 6 ijerph-19-14850-f006:**
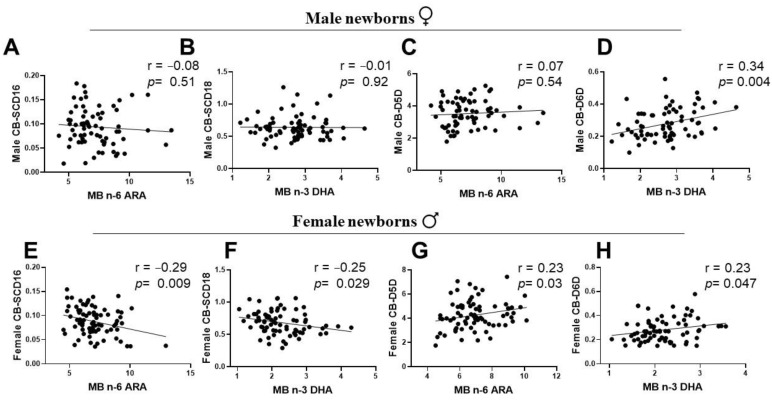
Maternal LC-PUFA negatively correlates with CB-SCD desaturation indices and positively with CB-PUFA desaturation indices (D5D and D6D) in female newborns. (**A**–**H**) Pearson relationship scatter plots between maternal n-3 and n-6 long chain derivatives of polyunsaturated unsaturated fatty acid (LC-PUFA) and cord blood desaturation indices in the newborn groups: (**A**–**D**) Males. (**E**–**H**) Females. Pearson *r* and *p* values are shown. LC-PUFA, long chain-polyunsaturated fatty acids; SCD, stearoyl CoA desaturase; D5D, delta-5-desaturase; D6D, delta-6-desaturase.

**Table 1 ijerph-19-14850-t001:** Baseline characteristics of pregnant south Indian women and newborns in the study.

Characteristics	Values	
	Male Newborns	Female Newborns
Maternal Gestational age (weeks)	39 ± 0.1	39 ± 0.3
Maternal age (years)	26 ± 4.2	26 ± 4.6
Number of new-borns, *n* = 153	*n* = 73	*n* = 80
Birth weight (kg)	3.04 ± 0.38	3.05 ± 0.36
Delivery method, *n*		
Spontaneous labor	48	55
Elective caesarean	25	25
Vegetarians	54	59
Non-habitual consumer of n-3 PUFA	19	21

Values are presented as mean ± standard deviation.

**Table 2 ijerph-19-14850-t002:** Fatty acid status (%) of plasma glycerophospholipids in maternal and umbilical cord blood.

Newborn Sex	Males		Females	
FA (%)	MB	CB	MB	CB
SFA				
C14(MA)	2.53 ± 1.54	3.39 ± 0.32 *	1.16 ± 0.49	0.75 ± 0.33 ^#^
C16(PA)	32.10 ± 2.17	29.1 ± 3.7 *	33.39 ± 2.1	31.84 ±2.23 ^#^
C18(SA)	7.797 ± 1.4	12.39 ±1.99 *	7.49 ± 1.19	12.93 ± 1.98 ^#^
MUFA				
C16:1n7(POA)	2.17 ± 1.01	2.82 ± 1.39 *	2.41 ± 0.71	2.78 ± 0.9 ^#^
C18:1n9(OA)	22.3 ± 3.1	7.81 ± 1.82 *	22.96 ± 3.4	8.92 ± 1.81 ^#^
PUFA n-6 family				
C18:2n6(LA)	18.44 ± 3.99	15.32 ± 3.1 *	19.59 ± 2.56	15.57 ± 2.61 ^#^
C20:3n6(DHGLA)	2.91 ± 0.81	4.05 ± 0.83 *	2.86 ± 0.97	4.25 ± 1.2 ^#^
C20:4n6(ARA)	7.21 ± 1.85	14.23± 3.17 *^$^	6.99 ± 1.54	16.96 ± 2.54 ^#$^
PUFA n-3 family				
C20:5n3(EPA)	0.26 ± 0.18	1.296 ± 1.11 *	0.01 ± 0.005	0.23 ± 0.06 ^#^
C22:5n3(DPA)	0.88 ± 0.34	1.36 ± 0.78 *	0.69 ± 0.39	0.84 ± 0.29 ^#^
C22:6n3(DHA)	2.75 ± 0.785	4.75 ± 1.40 *	2.25 ± 0.65	4.43 ± 1.25 ^#^
∑SFA	42.43 ± 2.69	46.83 ± 4.14 *	42.03 ± 2.56	45.53 ± 3.54 ^#^
∑MUFA	24.47 ± 2.78	10.63 ± 2.39 *	25.37 ± 3.07	11.71 ± 1.99 ^#^
∑PUFA	32.46 ± 2.96	41.02 ± 4.14 *	32.4 ± 2.75	42.26 ± 3.21 ^#^
∑n-6	28.56 ± 2.83	33.60 ± 2.86 *	29.44 ± 2.34	36.77 ± 2.8 ^#^
∑n-3	3.90 ± 1.1	7.41 ± 2.27 *	2.95 ± 0.84	5.49 ± 1.41 ^#^
∑n-6/n-3	7.96 ± 2.5	4.94 ± 1.49 *	10.61 ± 2.46	7.17 ± 2.03 ^#^
Desaturation Indices = Product/Precursor				
SCD16 index	0.07 ± 0.03	0.098 ± 0.05 *	0.072 ± 0.02	0.09 ± 0.03 ^#^
SCD18 index	2.93 ± 0.61	0.64 ± 0.17 *	3.14 ± 0.68	0.72 ± 0.24 ^#^
D5D index	2.58 ± 0.61	3.62 ± 0.99 *^$^	2.59 ± 0.71	4.27 ± 1.19 ^#$^
D6D index	0.18 ± 0.1	0.28 ± 0.09 *	0.12 ± 0.04	0.28 ± 0.09 ^#^

Values presented as mean ± SD. For compared groups MB vs CB of male new-borns, * indicates *p*-value < 0.05. For compared groups MB vs. CB of female new-borns, ^#^ indicates *p*-value < 0.05. ^$^ indicates *p*-value < 0.05 between CB of male and female newborns. Abbreviations: C14:0(MA), myristic acid; C16:0(PA), palmitic acid; C18:0(SA), stearic acid; C16:1n7(POA), palmitoleic acid; C18:1n9(OA), oleic acid; C18:2n6(LA), linoleic acid; C20:3n6(DHGLA), di-homo gamma linolenic acid; C20:4n6(ARA), arachidonic acid; C20:5n3(EPA), eicosapentaenoic acid; C22:5n3, docosapentaenoic acid; C22:6n3(DHA), docosahexaenoic acid.

## Data Availability

The data presented in this study are available on request from the corresponding author.

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
