# Peer review of "Maternal Plasma Glycerophospholipids LC-PUFA Levels Have a Sex-Specific Association with the Offspring’s Cord Plasma Glycerophospholipids-Fatty Acid Desaturation Indices at Birth"

_ijerph, 2022, doi:10.3390/ijerph192214850_

Round 1
Reviewer 1 Report
This study has potential to add to the evidence base, however, significant improvements are required. Please see suggestions below:
Introduction
Some improvements in grammar and wording/English needed
Lines 72-73: “Particularly in a maternal population dominated by vegetarians with no history of adequate n-3 LC-PUFA supplementation.” This sentence seems to be out of place/ needs further context or integrated into the previous or subsequent sentences
Lines 73-76: the aim is quite wordy. The second sentence of the aim starts with the word “And” which should not be done (sentences should not start with the word “and” – please amend and ensure the aim is clear, and that the content of the manuscript throughout is relevant to the aim.
Methods
Line 87-89: “A 24-hours recall method was used to assess dietary pattern of maternal subjects before delivery. The nutritional data obtained was recorded in a spread sheet. Dietary Intake 88 values (%) were generated based on Indian Food Composition Database (IFCT 2017).” – was the 24 hour recall completed by all participants at the same point in pregnancy e.g. the same trimester for each participant? What was the purpose of collecting information on dietary information for dietary patterns? Dietary patterns have not been mentioned elsewhere.
Figure 1: this seems a little disjointed – can the eligibility information/textbox and numbers be included underneath and in-line with. Also the line on the analysis of fatty acid composition seems out of place – I assume this follows from the n=73 and n=80 babies? The formatting of figure 1 would benefit from improvements.
Statistical analysis – confusing paragraph
Results
Line 145-146: “had an irregular intake of preformed n-3 DHA in their diet and a high n-6 LA as a source 145 of vegetable oil” – what is meant by “irregular intake of preformed n-3 DHA? For n-6 LA, do you mean vegetable oil was a source of n-6 LA, rather than n-6 LA a source of vegetable oil?
Line 147: first mention of an “oral survey” – what was this?
Line 147-148: “There was no significant difference in the dietary fat intake of compared maternal groups of male and female offspring” – confusing sentence
Table 1:
- There are two columns with values, but it is not clear what these columns show. Please amend
- What does “Sl. No.” in the first column on the left hand side mean?
- Source of PUFA intake – why only vegetable oils included here? Surely there were other sources, even if these contributed only small amounts to dietary intakes?
Fatty acid nomenclature is not consistent throughout – in some instances they are given as C18:2, in others the abbreviated letter format for FA e.g. ARA. Please amend so this is consistent throughout. It would be beneficial to give both the C18:2 and LA for linoleic acid, etc. when each fatty acid is first mentioned
Table 2:
- The layout of this table could be improved – the use of multiple symbols to indicate significance means there is a lot in this table and it is difficult to follow. Altering the layout of the columns and adding columns for p-values may improve readability
- This table shows fatty acid status % - this has not been included elsewhere in methods. Please include how fatty acid status will be expressed in the methods section
- Tables should standalone from text – please include the definitions for abbreviations in footnotes
- The information in Table 2 and supplementary table 2 appears to be the same – is there a reason this table has been included in both places?
Figure 3:
- The title/description under figure 3 uses letter abbreviations for the fatty acids (e.g., DHA), whereas the figure uses IUPAC format of C22:6n-3 etc. Please keep consistent throughout
Discussion
Lines 304-307: confusing sentence, please rewrite. Also, this sentence mentions the population is “dominated by vegetarians” – vegetarians have not been mentioned elsewhere in the manuscript. Please clarify what you mean here and add information to the manuscript as required. The phrase “dietary deficiency in n-3 LC-PUFA” is also unclear – what do you mean by this?
Lines 316-318: confusing sentences – please rewrite
Lines 338-344: Repetitive of results – discussion should elucidate findings, and compare to others
Overall, the discussion is difficult to follow in places, and would benefit from being made more succinct.
Author Response
Responses to Reviewer #1: This study has potential to add to the evidence base, however, significant improvements are required. Please see suggestions below:
Introduction
Point#1 Some improvements in grammar and wording/English needed
Lines 72-73: “Particularly in a maternal population dominated by vegetarians with no history of adequate n-3 LC-PUFA supplementation.” This sentence seems to be out of place/ needs further context or integrated into the previous or subsequent sentences.
Response: We have now rewritten the lines for better clarity.
Point#2 Lines 73-76: the aim is quite wordy. The second sentence of the aim starts with the word “And” which should not be done (sentences should not start with the word “and” – please amend and ensure the aim is clear, and that the content of the manuscript throughout is relevant to the aim.
Response: We have now rewritten the lines for better clarity.
Methods
Point#3 Line 87-89: “A 24-hours recall method was used to assess dietary pattern of maternal subjects before delivery. The nutritional data obtained was recorded in a spread sheet. Dietary Intake values (%) were generated based on Indian Food Composition Database (IFCT 2017).” – was the 24-hour recall completed by all participants at the same point in pregnancy e.g., the same trimester for each participant? What was the purpose of collecting information on dietary information for dietary patterns? Dietary patterns have not been mentioned elsewhere.
Response: Yes, an oral survey (24h recall) was completed before the delivery during ninth month of gestation period (last month of third trimester) for all the participants. We aimed to quantify the essential fatty acids and their metabolites LC-PUFA, which could interfere with study results of estimated desaturases activity in the MB and CB. So, it was important for us to know their sources i.e, dietary fat quality along with the blood analysis (relative information from the survey is mentioned in table 1 and supplementary table S1).
Point#4 Figure 1: this seems a little disjointed – can the eligibility information/textbox and numbers be included underneath and in line with. Also, the line on the analysis of fatty acid composition seems out of place – I assume this follows from the n=73 and n=80 babies? The formatting of figure 1 would benefit from improvements.
Response: Figure 1 is reformatted as per suggestion.
Point#5 Statistical analysis – confusing paragraph
Response: We have now rephrased the paragraph for more clarity.
Results
Point#6 Line 145-146: “had an irregular intake of preformed n-3 DHA in their diet and a high n-6 LA as a source of vegetable oil” – what is meant by “irregular intake of preformed n-3 DHA? For n6 LA, do you mean vegetable oil was a source of n-6 LA, rather than n-6 LA a source of vegetable oil?
Response: Irregular intake of preformed n-3 DHA in diet was written as some of the women consumed fish or DHA supplements but not on a regular basis. We are sorry for the typo in the line - ‘Vegetable oil was a source of high n-6 LA’- which is correct. We have now modified the text for more clarity.
Point#7 Line 147: first mention of an “oral survey” – what was this?
Response: We have now modified the methods to include the same. The oral survey (24h recall) was completed before the delivery during ninth month of gestation period (last month of third trimester) for all the participants.
Point#8 Line 147-148: “There was no significant difference in the dietary fat intake of compared maternal groups of male and female offspring” – confusing sentence
Response: We have now revised the sentence for better clarity.
Point#9 Table 1: - There are two columns with values, but it is not clear what these columns show. Please amend - What does “Sl. No.” in the first column on the left-hand side mean?
- Source of PUFA intake – why only vegetable oils included here? Surely there were other sources, even if these contributed only small amounts to dietary intakes?
Response: Table 1 is formatted for better presentation of data.
During oral diet survey, participants were not aware of other sources of PUFA. So, there is a greater possibility of consumption other PUFA sources is negligible. The only source of PUFA common between compared groups in this study population was vegetable oil rich in n-6 linoleic acid (sunflower oil) for both vegetarians and non-vegetarians. Also, our observation of diet quality is in line with previous studies (references mentioned below). As we are not aware of other sources, we have removed this parameter from the table.
References
- Mani I, Kurpad AV. Fats & fatty acids in Indian diets: Time for serious introspection. Indian J Med Res. 2016;144(4):507-514. doi:10.4103/0971-5916.200904
- Mani I, Dwarkanath P, Thomas T, Thomas A, Kurpad AV. Maternal fat and fatty acid intake and birth outcomes in a South Indian population. Int J Epidemiol. 2016 Apr;45(2):523-31. doi: 10.1093/ije/dyw010. Epub 2016 Mar 24. PMID: 27013336.
Point#10 Fatty acid nomenclature is not consistent throughout – in some instances they are given as C18:2, in others the abbreviated letter format for FA e.g., ARA. Please amend so this is consistent throughout. It would be beneficial to give both the C18:2 and LA for linoleic acid, etc. when each fatty acid is first mentioned
Table 2: - The layout of this table could be improved – the use of multiple symbols to indicate significance means there is a lot in this table and it is difficult to follow. Altering the layout of the columns and adding columns for p-values may improve readability
- This table shows fatty acid status % - this has not been included elsewhere in methods. Please include how fatty acid status will be expressed in the methods section
- Tables should stand alone from text – please include the definitions for abbreviations in footnotes - The information in Table 2 and supplementary table 2 appears to be the same – is there a reason this table has been included in both places?
Response: We have now made the nomenclature of fatty acids consistent in the manuscript.
Layout of table 2 is improved and required information is added in methods. We did not add p-value as it could confuse the readers, as there are three pairs of comparison between the groups (MB versus CB of male newborns, MB versus CB of female newborns and CB of male versus CB of female newborns).
Addition suppl. Table 2 was a repetition by mistake and therefore, it has been removed from suppl. files.
Point#11 Figure 3: - The title/description under figure 3 uses letter abbreviations for the fatty acids (e.g., DHA), whereas the figure uses IUPAC format of C22:6n-3 etc. Please keep consistent throughout
Response: We have now made the nomenclature consistent in figure 3 and have given the details in the footnotes.
Discussion
Point#12 Lines 304-307: confusing sentence, please rewrite. Also, this sentence mentions the population is “dominated by vegetarians” – vegetarians have not been mentioned elsewhere in the manuscript. Please clarify what you mean here and add information to the manuscript as required. The phrase “dietary deficiency in n-3 LC-PUFA” is also unclear – what do you mean by this?
Response: Table 1 is now modified with supplementary data information submitted previously about the number of vegetarians in the study. Of note, the study population were not getting enough n-3 DHA through diet/supplementation, because they were mainly vegetarians. We have now revised the sentences for more clarity.
Point#13 Lines 316-318: confusing sentences – please rewrite
Response: The above-mentioned lines are rewritten for clarity.
Point#14 Lines 338-344: Repetitive of results – discussion should elucidate findings, and compare to others Overall, the discussion is difficult to follow in places, and would benefit from being made more succinct.
Response: We have now edited discussion to deliver concise results of the study with follow up on current literature available.
Reviewer 2 Report
This is an extensive manuscript, with very interesting conclusions, regarding fatty acid metabolism, and the links between maternal and fetal metabolism. The comments I wish to make are mainly linked to the manner the data are presented and not with the data itself.
Strengths
This is an ample study on the Developmental Origins of Health and Diseases, well documented and well written. The data are well analyzed, and the valid points are referenced accordingly.
Weaknesses
The article is extremely extensive and while that’s generally a good thing, in this context, all the data that are presented can take away from the general conclusion, i.e., the gender-differentiation of fatty acids metabolism.
Minor points
The authors should turn off the automated end-of-row word-breaking feature, as it can be a bit troublesome (including in the title).
Due to the peculiarities of the analyzed population, I would remove the last row in Table 1 (Maternal tobacco/alcohol exposure) – it is implied, and replace it with the Maternal dietary regimen (vegetarian or not) from the Supplementary files. In the same table, I would differentiate male and female infants in the first line of the table.
I would be interested in what the authors consider a “full-term” pregnancy. According to the World Health Organization, I believe it is between 39 and 40 completed weeks of gestation, and in this respect, I don’t see how the mean GA for both males and females can be 39 ± 0.1 and 39 ± 0.3 weeks, respectively.
In the Exclusion criteria, the authors mention physiological jaundice (row 97) – I would be interested in why and how – was it done retrospectively, meaning the blood was first collected, then excluded from the analysis? Were the subjects aware of it being a possibility? Also “cases of anemia” – was it maternal anemia? This should be more precisely written.
On row 128, replace “glaser” with “Glaser”.
Author Response
Reviewer #2: This is an extensive manuscript, with very interesting conclusions, regarding fatty acid metabolism, and the links between maternal and fetal metabolism. The comments I wish to make are mainly linked to the manner the data are presented and not with the data itself.
Strengths: This is an ample study on the Developmental Origins of Health and Diseases, well documented and well written. The data are well analysed, and the valid points are referenced accordingly.
Weaknesses: The article is extremely extensive and while that’s generally a good thing, in this context, all the data that are presented can take away from the general conclusion, i.e., the gender-differentiation of fatty acids metabolism.
Minor points:
Point#1 The authors should turn off the automated end-of-row word-breaking feature, as it can be a bit troublesome (including in the title).
Response: Thank you for the comment. We have now turned off the feature.
Point#2 Due to the peculiarities of the analysed population, I would remove the last row in Table 1 (Maternal tobacco/alcohol exposure) – it is implied, and replace it with the Maternal dietary regimen (vegetarian or not) from the Supplementary files. In the same table, I would differentiate male and female infants in the first line of the table.
Response: Table 1 is modified with above mentioned constructive suggestions.
Point#3 I would be interested in what the authors consider a “full-term” pregnancy. According to the World Health Organization, I believe it is between 39 and 40 completed weeks of gestation, and in this respect, I don’t see how the mean GA for both males and females can be 39 ± 0.1 and 39 ± 0.3 weeks, respectively.
Response: Gestation in singleton term pregnancies lasts an average of 40 weeks (280 days). We considered pregnancies between 270-280days of gestation period as ‘term’. We converted days to weeks from the clinical case history records of the hospital and estimated the gestation period of the participants.
For example: A mean of 274 days is 39.1 weeks [274days/7days (week)] and mean of 275 days is 39.3 weeks.
Point#4 In the Exclusion criteria, the authors mention physiological jaundice (row 97) – I would be interested in why and how – was it done retrospectively, meaning the blood was first collected, then excluded from the analysis? Were the subjects aware of it being a possibility? Also “cases of anaemia” – was its maternal anaemia? This should be more precisely written.
Response: Cases of physiological jaundice in newborns, maternal anaemia were excluded. The samples were first collected then excluded according to the exclusion criteria in view of potential risks to sampling errors.
These conditions are known to interfere with the lipid metabolism (references below), and thus affect the blood fatty acid composition examined in the study.
The subjects were made aware of this as a possibility.
We have now specified maternal anaemia in the manuscript.
References
- Rees WD, Hay SM, Hayes HE, Stevens VJ, Gambling L, McArdle HJ. Iron deficiency during pregnancy and lactation modifies the fatty acid composition of the brain of neonatal rats. J Dev Orig Health Dis. 2020 Jun;11(3):264-272. doi: 10.1017/S2040174419000552
- Shibuya, A., Itoh, T., Tukey, R. et al. Impact of fatty acids on human UDP-glucuronosyltransferase 1A1 activity and its expression in neonatal hyperbilirubinemia. Sci Rep 3, 2903 (2013). https://doi.org/10.1038/srep02903
Point#5 On row 128, replace “glaser” with “Glaser”.
Response: Replaced glaser with ‘G’ capital
Reviewer 3 Report
Dear Authors
It is nice to read a well constructed manuscript for a change so thank you. I only have minor comments
1. Can you please check that all abbreviations are fully worded in first usage. You do cover all in the list of abbreviations but feel there are several that don't have first usage in full name
2. Any chance of describing the cord blood as arterial or venous as this may impact your results, especially with respect to the desaturase conclusions
3. There are a few layout and capitals to deal with but the editorial team will do a better job than me
4. Could the CB-D5Dindex in female new borns be a type 1 error statistically?
Author Response
Reviewer #3: Dear Authors It is nice to read a well-constructed manuscript for a change so thank you.
I only have minor comments
Point#1 Can you please check that all abbreviations are fully worded in first usage. You do cover all in the list of abbreviations but feel there are several that don't have first usage in full name
Response: We have thoroughly checked the manuscript and added the fully worded abbreviations in first usage, followed by abbreviations in the latter texts.
Point#2 Any chance of describing the cord blood as arterial or venous as this may impact your results, especially with respect to the desaturase conclusions
Response: We have now mentioned the use of venous cord blood in the study throughout the manuscript.
Point#3 There are a few layouts and capitals to deal with but the editorial team will do a better job than me. Could the CB-D5Dindex in female newborns be a type 1 error statistically?
Response: We have revised the manuscript as suggested. Yes, it could be type I error statistically.
Reviewer #3: Dear Authors It is nice to read a well-constructed manuscript for a change so thank you.
I only have minor comments
Point#1 Can you please check that all abbreviations are fully worded in first usage. You do cover all in the list of abbreviations but feel there are several that don't have first usage in full name
Response: We have thoroughly checked the manuscript and added the fully worded abbreviations in first usage, followed by abbreviations in the latter texts.
Point#2 Any chance of describing the cord blood as arterial or venous as this may impact your results, especially with respect to the desaturase conclusions
Response: We have now mentioned the use of venous cord blood in the study throughout the manuscript.
Point#3 There are a few layouts and capitals to deal with but the editorial team will do a better job than me. Could the CB-D5Dindex in female newborns be a type 1 error statistically?
Response: We have revised the manuscript as suggested. Yes, it could be type I error statistically.
Round 2
Reviewer 1 Report
The responses to reviewer comments were difficult to follow. In the future please provide line numbers or an indication of where the text has been changed. The methods needs to contain information on all data collected. Please ensure all information is included here. There is a lot of analysis included in the manuscript which is difficult to follow at times.
Author response to round 1 revisions: “an oral survey (24h recall) was completed before the delivery during ninth month of gestation period (last month of third trimester) for all the participants” - Does this mean in last month of pregnancy? Not overly clear here.
Line 89 – “a 24-hours recall method for oral survey” – please reword
Line 91 – qualitative information? What qualitative information was collected and how?
Figure 1 - the line on the analysis of fatty acid composition still seems out of place
“Irregular intake of preformed n-3 DHA in diet was written as some of the women consumed fish or DHA supplements but not on a regular basis” – Not habitual consumers of PUFA would be more reflective of this
Table 1 - What are the numbers in the first column? Are these needed?
Author response to round 1 revisions: “During oral diet survey, participants were not aware of other sources of PUFA” - I am confused about what is meant by “participants were not aware of other sources of PUFA”. For the 24 hour recall were participants asked to recall all foods they consumed in a 24 hour period? Were the participants specifically asked what foods they consumed contained PUFA? If participants were asked which PUFA rich foods they consumed this would introduce considerable issues such as bias as this will be very dependent on participant knowledge of PUFA. Please clarify.
Author response to round 1 revisions: “Table 1 is now modified with supplementary data information submitted previously about the number of vegetarians in the study. Of note, the study population were not getting enough n-3 DHA through diet/supplementation, because they were mainly vegetarians. We have now revised the sentences for more clarity” - This needs further discussion and justification in the manuscript. Recommendations for dietary intake of PUFA are not included in the manuscript, but this information is necessary to give this context.
Table S1 – nutritional questionnaire results – how was this data collected?? What questionnaire?
Author Response
Point 1. The responses to reviewer comments were difficult to follow. In the future, please provide line numbers or an indication of where the text has been changed. The methods need to contain information on all data collected. Please ensure all information is included here. There is a lot of analysis included in the manuscript which is difficult to follow at times.
Response: We are sorry about the inconvenience. Now, we have highlighted the font in red color to indicate modifications. We have now modified section 2.1 (in red) to include more information about the data collected.
Point 2. Author response to round 1 revisions: “an oral survey (24h recall) was completed before the delivery during ninth month of gestation period (last month of third trimester) for all the participants” - Does this mean in last month of pregnancy? Not overly clear here.
Response: Yes, the oral survey was completed in the last month of pregnancy. We have revised the sentence in section 2.1 for more clarity. “An oral survey using 24-hours recall method was done to assess the dietary pattern of maternal subjects before delivery (last month of the pregnancy).”
Point 3. Line 89 – “a 24-hours recall method for oral survey” – please reword
Response: The above-mentioned line is rewritten and highlighted in the manuscript. “An oral survey using 24-hours recall method was done to assess the dietary pattern of maternal subjects before delivery (last month of the pregnancy).”
Point 4. Line 91 – qualitative information? What qualitative information was collected and how?
Response: ‘Qualitative information’ mentioned in the manuscript is a part of 24hour recall. Because the information received from study participants was about -what they consumed in the last 24 hours. This data did not provide the absolute quantity of food consumed. Hence, this data was termed as qualitative data/information. We have now elaborated on what type of information was collected in section 2.1.
Point 5. Figure 1 - the line on the analysis of fatty acid composition still seems out of place
Response: We have now modified Figure 1 as per indication.
Point 6. Irregular intake of preformed n-3 DHA in diet was written as some of the women consumed fish or DHA supplements but not on a regular basis” – Not habitual consumers of PUFA would be more reflective of this
Response: We have added the line – “Not habitual consumers of n-3 DHA PUFA” in table 1 and section 3.1 (highlighted in red).
Point 7. Table 1 - What are the numbers in the first column? Are these needed?
Response: We have now removed the first column (numbers) from Table 1.
Point 8. Author response to round 1 revisions: “During oral diet survey, participants were not aware of other sources of PUFA” - I am confused about what is meant by “participants were not aware of other sources of PUFA”. For the 24 hour recall were participants asked to recall all foods they consumed in a 24 hour period? Were the participants specifically asked what foods they consumed contained PUFA? If participants were asked which PUFA rich foods they consumed this would introduce considerable issues such as bias as this will be very dependent on participant knowledge of PUFA. Please clarify.
Response: In the 24h recall survey (oral survey), participants were asked to recall foods they consumed in the last 24h. Then, they were asked about their diet during pregnancy in general. They were not specifically asked for PUFA containing foods. We are extremely sorry for the confusion. Writing “participants were not aware of other sources of PUFA” was an error from our side.
Point 9. Author response to round 1 revisions: “Table 1 is now modified with supplementary data information submitted previously about the number of vegetarians in the study. Of note, the study population were not getting enough n-3 DHA through diet/supplementation, because they were mainly vegetarians. We have now revised the sentences for more clarity” - This needs further discussion and justification in the manuscript. Recommendations for dietary intake of PUFA are not included in the manuscript, but this information is necessary to give this context.
Response: Recommendations for dietary PUFA intake context is now added in the introduction part of manuscript (para 1, highlighted in red). Moreover, we have now modified the discussion to add this as a limitation (para 6&7, highlighted in red).
Point 10. Table S1 – nutritional questionnaire results – how was this data collected?? What questionnaire?
Response: Nutritional questionnaire was collected by oral survey (24h recall questionnaire method) by qualitative method and then, relatively quantified by Indian Food Composition Database (IFCT), 2017. We have now modified section 2.1 to include these information (highlighted in red)